# Discovering the Legacy of Hispanic/Spanish and South American Landscapes through Geohistorical Sources: The Geographical and Topographical Relations of Philip II

**Alejandro Vallina-Rodríguez** [1],*[ID]**, Ángel I. Aguilar-Cuesta** [2][ID]**, Laura García-Juan** [3]**, Miguel B. Bernabé-Crespo** [3][ID]**, Miguel A. Bringas-Gutiérrez** [4]**and Concepción Camarero-Bullón** [3],*

1  Department of Geography, University of Salamanca, 00503 Ávila, Spain
2  Department of Social Sciences, Valencian International University, 46002 Valencia, Spain; aaguilarc@universidadviu.com
3  Department of Geography, Autonomous University of Madrid, 28049 Madrid, Spain; laura.garciaj@uam.es (L.G.-J.); miguelb.bernabe@uam.es (M.B.B.-C.)
4  Department of Economics, University of Cantabria, 39006 Santander, Spain; miguel.bringas@unican.es
*  Correspondence: alvallina@usal.es (A.V.-R.); concepcion.camarero@uam.es (C.C.-B.); Tel.: +34-(663)-179812 (A.V.-R.)

**Abstract:** Landscapes have history and memory, which are eloquent generators of testimonies and traces on the processes of the landscape that take place today, and that will take place in the future. In recent years, numerous methods of analysing land and landscape patterns have been developed and evaluated, based on the multiplicity of these type of geographic and historical data sources, which have developed the concept of the geohistorical source. The goal of these sources of information allows us to historically reconstruct landscapes. With this in mind, the basic objective of the present research is to approach a geohistorical source with a wide spatial spectrum in Europe and America: the geographical and topographical relations of Philip II. This source has been chosen for the quality, quantity, variety and systematization of the data it provides on the territory and landscape of the crown of Castile. In addition, it ended up being the model of how to obtain organized and homogeneous knowledge of a large spatial area, considering the geographical, anthropological and historical data of the different territories. This geohistorical source is reliable, because the local authorities, both secular and ecclesiastical, are questioned, as they are the ones who inhabit, use, and, at different levels, govern the territory and its people.

**Keywords:** geohistorical source; historical geography; territory; landscape

## 1. Introduction

Our current perspective of landscapes and terrains is the result of their evolution over time. For this reason, and as a means to understand the characteristics of a given historical place and society, researchers must consult documents, references and data that provide information about societies, and the lands in which they occupy and place value. Historical geography, in addition to other branches of knowledge such as history, economics, anthropology and medicine, among others, use specific features or characteristics to form a corpus of information that allows reliable interpretation of the past [1] (p. 142). To this end, spatial analytical approaches have traditionally been used, employing a wide range of textual resources, cadastral or paracadastral documents, cartographic materials, photographs, statistics and censuses and literary resources, which have been increasingly combined with technologies such as geographic information systems or, more recently, Big Data [2] (p. 45). Thus, in recent years, numerous methods for analysing land and landscape patterns have been developed and assessed, based on a multitude of geographical and historical data sources, and have given rise to the concept of a "geohistorical source" [3]

(p. 20). Under the umbrella of this notion is a set of sources containing geographical and historical data which can be spatialised [4] (p. 69).

The landscape, aside from the features that any geographical space can acquire, is an instrument that shapes how reality is perceived and helps in the interpretation and understanding of a particular reality within a specific spatial area. Hence, approaches involving the perspective of time require the analysis of all available features and characteristics, with the premise that in order to understand the present and be able to foresee the future, it is necessary to know how to interpret the past histories of landscapes, which, on many occasions, constitute a rich source of information regarding the structural changes that have taken place, as well as the origins of the socio-cultural perceptions of the inhabitants [5] (p. 210). It must therefore be stated that for modern geography [6] (p. 98), as in other sciences and branches of knowledge that focus on societies and the environment, landscapes are the holders of both history and memories. They are the eloquent creators of clues and traces of the landscape processes that are currently taking place and those that will take place in the future [7] (p. 23).

The key to all of this is to arrive at the historical reconstruction of landscapes, which is undoubtedly linked to sources of information, how to find and deal with these sources, the essential information and circumstances they provide, and the interpretation and discussion of the data. In light of the above, it is extremely important to identify the right sources of information for each particular purpose and to read and interpret the background and references correctly. This exercise on environmental history, of which only the view from human geography is presented here, is not always easy to achieve, mainly because finding the necessary data to reconstruct the landscapes of the recent past becomes, on most occasions, an arduous task and is not exempt to difficulties. Geohistorical sources contain a huge amount of data which, if well managed, provide extensive information about the structure of societies at the time they were created [8] (p. 171). However, these volumes present a vast amount of diverse and varied information that often impedes their handling, owing to the wide learning curve required in order to use the sources in which they appear. For the purpose of clarifying, as much as possible, the basic typologies of geohistorical sources that can be used in the understanding and the study of landscapes, the following categorisation based on different criteria, which in no way excludes other approaches, is proposed:

- Bibliographic sources: These are understood, suitably, as written sources, generally in printed form, and in particular literature of a diverse nature and production, in which information on landscapes appears in a tangential way or can be the object of interpretation by the reader. The data of this type of documentation in the field of landscapes, although difficult to find, are quite relevant with regard to epistolary relationships, literature and travel accounts, encyclopaedias and local historiographies, among other texts suitable for studying landscapes, terrains and societies.
- Cartographic and iconographic sources: The sets of maps and plans, as well as other products linked to these, such as planimetric sketches, sketches, views and drawings and historical photographs, lithographs, drawings, paintings, watercolours, etc., not only allow an objective approach to territorial phenomena, but also allow subjective perspectives to be drawn about the reality of rural and urban areas [9] (p. 11).
- Direct textual sources: This typology of geohistorical sources is probably the most interesting and relevant in terms of the volume and quality of historical information they provide on landscapes and their components. In a rather summarised form, it can be stated that a large part of this typology derives from fieldwork, i.e., it derives to a greater or lesser degree from the inspections carried out in the territories in question. These documents, which include land registers, topographical dictionaries, reports and studies of the period, replies to interviews and questionnaires and other similar elements, require a great mastery of the subject to be analysed. This also includes a detailed understanding of the study area, the document itself and its authors, and the

use of a clear scientific method in order to extract the information required for the type of study to be carried out (Figure 1).

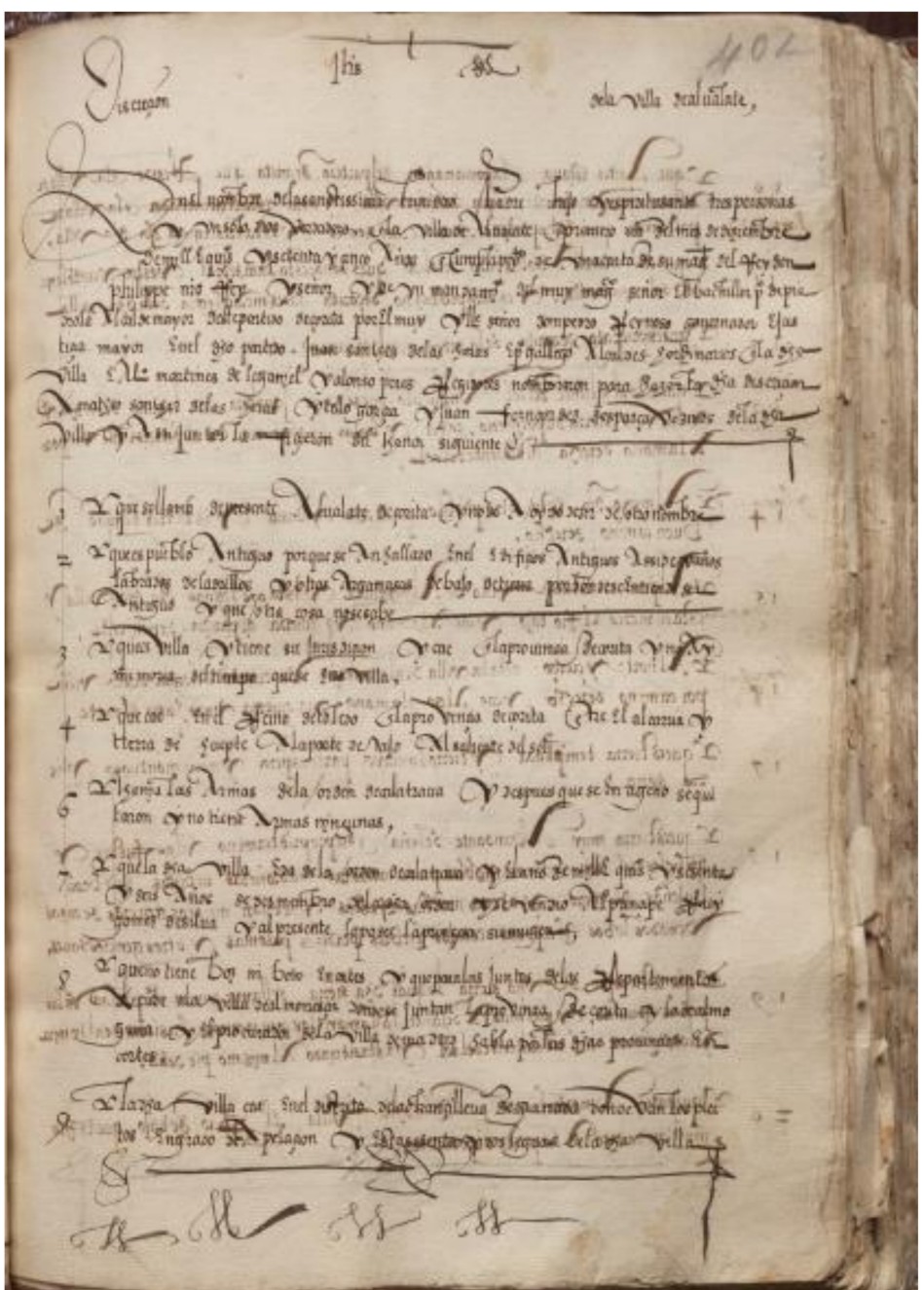

**Figure 1.** Extract from the Topographic Relation of Philip II for the municipality of Albalate de Zorita (Guadalajara, Spain). Library of the Royal Monastery of El Escorial. Manuscript, paper 318 × 215 mm. Signature J-I-15 [10]. Reproduced from [or Adapted from] "Relationships of peoples of Spain made in the time of Philip II, for the years from 1574 to 1580" by the authors of the research (https://rbmecat.patrimonionacional.es/cgi-bin/koha/opac-detail.pl?biblionumber=695 (accessed on 24 October 2021)). Public Domain Work.

- Indirect textual sources: Although normally less comprehensive than the previous sources, a set of information on local reality can be derived indirectly from this typology. From historical documentation such as medical topographies, ecclesiastical rogations, lawsuits and the collection of tariffs and duties, memorials of complaints, etc., it

is possible to extract information to be considered in relation to the history of a landscape, especially in the absence of other more exact direct documentary sources [11] (p. 13). Research using these types of sources must be carried out in a scientific manner. Additionally, the information obtained must be compared, analysed and completed with that provided by other available sources, allowing for a correct, careful and logical analysis.

In this initial approach to geohistorical sources as the basis for the study of landscapes, it is necessary to bear in mind that they are heterogeneous but complementary sets of sources. Their interest, therefore, lies in the information they contain, not only on aspects highly related to land use, but also in the testimony they provide on the relationship between these aspects and the physical, biological, historical, social, environmental or demographic facts of the geographical frameworks to which they refer.

## 2. Materials and Methods: Geographical Examinations as a Source for Landscape Study and Reconstruction

The increase in the number of studies on landscapes has led to a significant rise in the use of all kinds of geohistorical sources, sometimes in the absence of the knowledge required to understand the source being consulted [12] (p. 17). In order to understand a source, extracting the information it offers the researcher, it is necessary to answer the following questions [13] (p. 11). What is the purpose of the source? What type of information does it contain and with what criteria was the data collected, structured and prepared? What techniques were used to collect the information, to ensure its accuracy and correctness, and to prepare the data? In what political, social and economic context was the assessment carried out? Who was or who were those ultimately responsible for the source? Consequently, this collection of reflections, as well as a few more, can be summarised in four main questions that all researchers must ask themselves when confronted with any source to be used, but especially with those of a geo-historical nature, given the length in time since the sources were prepared [14] (p. 171):

- What is the socio-economic, political and technical context in which the source was developed and carried out?
- Why was it carried out?
- What does it involve?
- Of what use is it currently to the researcher?

It should be noted that this lack of knowledge, which is more widespread than desirable, has all too often led to somewhat surprising conclusions and has highlighted the need to undertake a systematic and in-depth study of the geo-historical sources themselves before proceeding to the excavation and interpretation of the data they contain. This has led to a line of research in which much progress is already being made, especially in the case of cadastral and paracadastral sources.

Specifically, the research will base its research criteria on the hypothetico-deductive method, one of the most used models in the geographical sciences and humanities. According to this method, it has been established as a fundamental hypothesis that the landscape and the territory have a clear reference of knowledge in geohistorical sources. Specifically, examples of the Topographic Relationships of Philip II are used to observe the knowledge of the landscape that was had in the 16th century in America and Europe. From this, the phenomena included in the geographical descriptions are explained, the most elementary consequences or implications of the hypothesis itself are deduced, and, finally, the deduced statements are checked or refuted by comparing them with the experience and current state of the territory.

Geographical or geographical-like surveys, carried out in the interest of becoming familiar with the territory or for other purposes, are a widely used set of sources due to the immediacy of the data they provide and the ease with which they can be processed and interpreted [15] (p. 12). Ease, it must be said, is more apparent than real. In the following pages we describe an approach, involving important surveys of spaces on both

sides of the Atlantic, namely, the Geographical and Topographical Relations of Philip II. This source was chosen because of the quality, quantity, variety and systematicity of the data it provides. In addition, it has become the model to follow for obtaining organised, systematic and homogeneous information about an immense space, without eliminating the geographical, anthropological and historical specificities of the different territories. Owing to space limitations, we have left out other key surveys carried out during 18th century Spain, of which the Philippine survey is most notable and is referred to in the following works: ([16] (p. 67); [17] (p. 93); [18] (p. 26); [19] (p. 110); [20] (p. 47); [21] (p. 42)).

## 3. Results

The 16th century marked the beginning of the efforts of European monarchies to design tools and methods to obtain territorial information that, when suitably performed, would allow them to systematically survey their lands and the people inhabiting the area. In Spain, such practices were initiated by Philip II (1527–1598), who was king of Spain and the Indies from 1554 and of Portugal from 1580. King Philip II is the monarch Richard Kagan refers to as "the king among geographers" [22] (p. 34), and who has also been referred to as the *rey papelero* or "the paper king". As far as we are concerned, here we are dealing with a Renaissance prince, who, before acceding to the throne, had travelled around Europe, embraced the values of humanism, and, who, following the abdication of his father, Charles I of Spain and V of the Holy Roman Empire, came to rule over territories on which the sun never sets. This was a varied and diverse territory, one that the king wanted to know and was obliged to know and one that was to be described in texts and mapped out. Philip II chose to stop living for the Empire, as his father and the war against Protestantism had done. He then focused his attention on the Hispanic territories, including those inherited by the dynasty, the Netherlands, a key area for the development of the cartography of the time, and the territories of the Spanish Empire. Philip II was also the king who was responsible for the construction of El Escorial, which was to become the pantheon of the dynasty, a monastery, a royal site and a centre for study. El Escorial was endowed with a magnificent library for which he had acquired all types of books, manuscripts and maps, regardless of their provenance and content.

For America, the reign of Philip II signified what has come to be called "the new stage of Hispanic America", giving way to the pacification of the territory and its organisation, after the discovery and conquest of the Americas. This was the time of the founding of extremely important cities: Mexico, Santafé, Cartagena de Indias, Lima, Santiago de Chile, Buenos Aires, among others (Figure 2).

In this state of affairs, the Casa de Contratación and the reorganization of the Council of the Indies, based on new ordinances approved in September 1571, would be key to the task of getting to know this large and now consolidated territory. These defined the post of the Major Chronicler of the Indies and that of the cosmographer. The former was entrusted with creating an archive in which all the documents relating to America were to be kept, while the task of the latter position was to write a general geography of the Indies, based on the particular reports sent to him from the provinces. Initially, both positions converged with one person being appointed, Juan López de Velasco, who collaborated with Juan de Ovando, President of the Council of the Indies between 1570 and 1774. Together they drafted the questionnaires that were sent to America, dated 1569, 1571 and 1577, organised the scarce amount of material that was gathered, and wrote the "Geography and General Description of the Indies" (1571–1575), based on the reports received and the documentation that was being gathered from the New World.

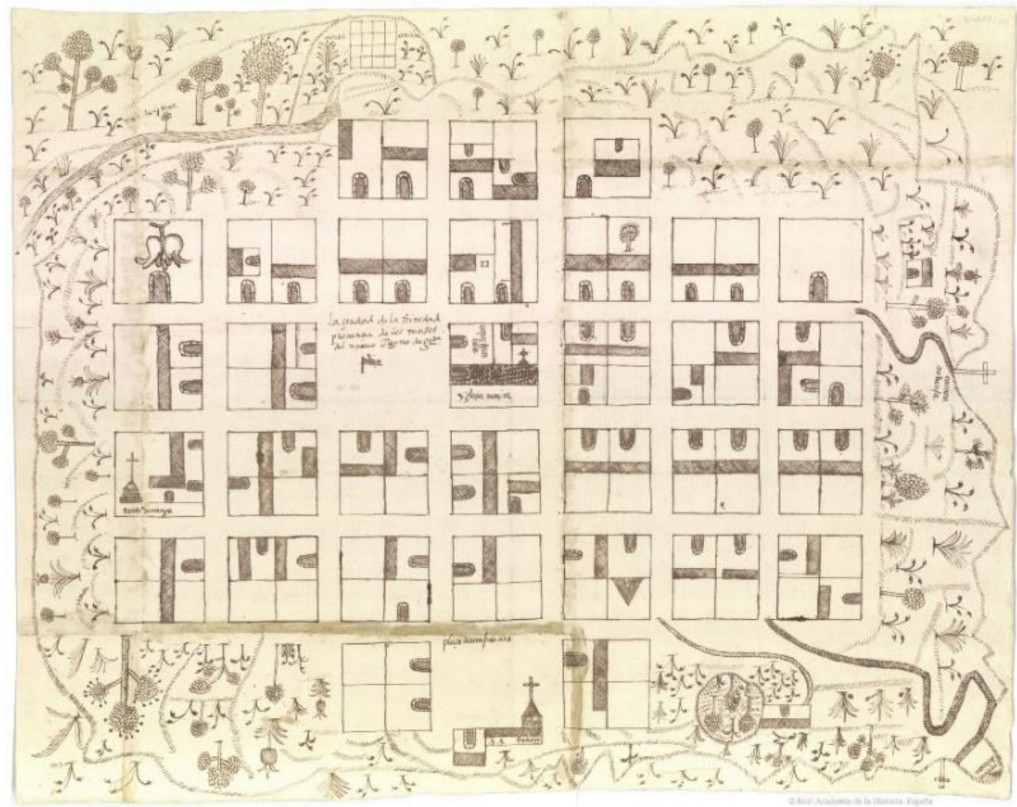

**Figure 2.** The city of Trinidad, province of the Musos in the New Kingdom of Granada. Library of Royal Academy of the History of Spain. Map: 54 × 69 cm. Collection: Cartography and Graphic Arts Section-Signature: C-028-002 [23]. Reproduced from [or Adapted from] Royal Academy of History Digital Library by the authors of the research (https://bibliotecadigital.rah.es/es/catalogo_imagenes/grupo.do?path=1049537 (accessed on 24 October 2021)). Public Domain Work.

Nonetheless, it was not only necessary to obtain information about America, but it was also necessary to explore Spain. To this end, a procedure was carried out which involved asking the local authorities on both sides of the sea to answer a survey or questionnaire regarding the territory and its inhabitants (in reality, several surveys were sent out, but with the same basic information). These questionnaires were dispersed mainly in the 1570s, but not exclusively. As we have seen, the questionnaires, although basically the same, had specific aspects concerning each of the territories. The information requested was structured in a series of points including the name of the place (current and former), type of jurisdiction, geographical situation, topographical characteristics, river courses, distance to surrounding areas, administrative organisation, demographic aspects (population and settlement), natural resources, tax burdens, municipal and council resources, aspects of town planning, buildings, agricultural, livestock and forestry activities, craft and industrial activities, mining (mines, quarries, salt mines, etc.), liberal ideas, services, transport, aspects of health and welfare, public and private education, religious aspects, and miscellaneous subjects such as history, legends, art among others. In the case of America, in addition to the abovementioned topics, specific information was requested about the environment: the ancient name of the population in the indigenous language or languages and its meaning, indigenous toponymy and its meaning, language spoken, ethnicity, legends and historical facts, mountains, rivers and coasts, tides, and islands, etc. It was undoubtedly known that place names were valuable for gaining insight into the elements comprising the landscape and the resources of villages, their location and their history [24] (p. 72) (Figure 3).

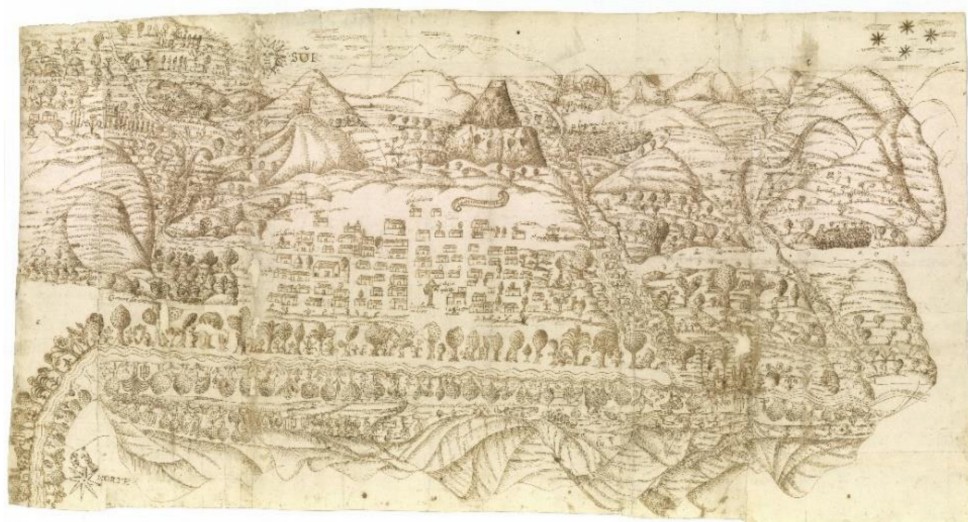

**Figure 3.** The city of Toquio, 28 January 1579. Library of Royal Academy of the History of Spain [C-028-001]. El Tocuyo (Venezuela) [25]. Reproduced from [or Adapted from] Royal Academy of History Digital Library by the authors of the research (https://bibliotecadigital.rah.es/es/consulta/resultados_ocr.do?id=1579&forma=ficha&tipoResultados=BIB&posicion=5 (accessed on 24 October 2021)). Public Domain Work.

This research will briefly refer to toponymy, as this concept reflects the location of the localities, the elements of the surrounding landscape and the conjunction with historical-legendary facts. In particular, these facts are based on the relationship of the town of Segura de la Sierra (Jaén, Spain), located at the top of the Sierra of the same name, in the shadow of a castle, in a territory that, for several centuries, was the border between the lands of Castile and the Nasrid Kingdom. In relation, reference is made to an earlier place name, Altamira, which is directly related to the altitude of its location and the territory in sight, and to the second, which reflects the idea of altitude from the Sierra and the idea of security from the enemy, with the name Segura (safe). It also refers to a time when the locality was more important, as it seems to have had the status of a city, which was lost in the 16th century, probably when the strategic value of this territory disappeared once the Reconquest ended with the conquest of the Nasrid Kingdom of Granada in 1492.

This town is now called Segura de la Sierra, and in the past, it used to be a city called the city of Altamira, and thus these mountains were called the lands of Altamira. It is said that a King came fleeing and took refuge in its fortress, which was very high in such a way that it almost seems that in some parts he was safe from everyone; he said "here, I am safe". It is said that the name, Segura, was taken from this account.

In addition to the textual information, resulting from the answers provided in the survey, it was requested that a sketch be made of the locality or area to which the report referred, sometimes involving larger bodies of land in the case of America. However, in the case of Spain, the sketches were always of local areas. Consequently, there is now a set of maps that constitute a collection of non-technological cartography, which could be called "popular", of great interest. This is owed to the information this source contains and because it reflects the perception of space and the elements of the landscape of those who populated it, answered the questionnaire, and sketched the area. To a large extent, this information provides an account of the natural elements of the landscape, in terms of the characteristics of the geoforms contained within the territory, the natural vegetation and the anthropic use of the soil, and, finally, the recollection of the fluvial resources available in the territory assayed. Additionally, this type of information is of enormous value for understanding the terrain and the landscape of the areas surveyed, as it contains reports that, to a greater or lesser degree, provide accounts of the main consubstantial and visual aspects of the landscape of the epoch [26] (p. 41) (Figure 4).

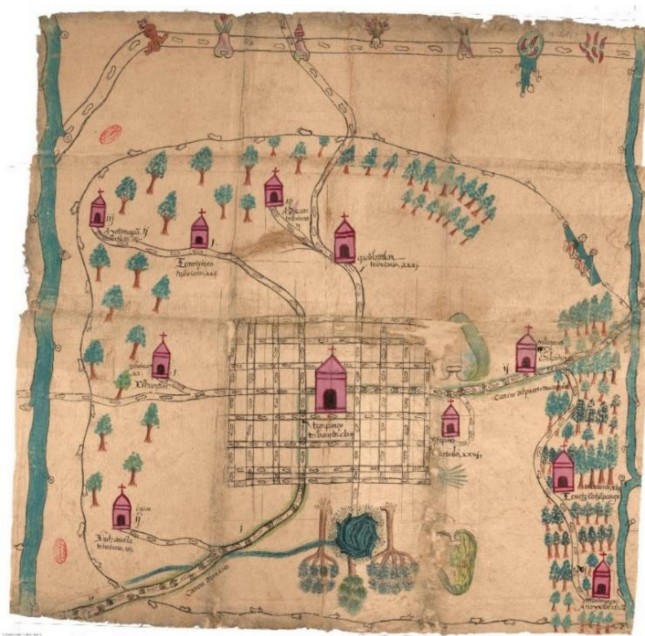

**Figure 4.** Map of Minas de Zumpango, Archbishopric of Mexico, 09-04663, nº 36, 10 March 1582. Library of Royal Academy of the History of Spain [27]. Reproduced from [or Adapted from] Royal Academy of History Digital Library by the authors of the research (https://bibliotecadigital. rah.es/es/consulta/registro.do?control=RAH20110000394 (accessed on 24 October 2021)). Public Domain Work.

It is important to note that, in terms of graphs, the wealth of information comprising the Indies Relations is incomparably superior to that of the Spanish Relations. Of these, only two sketches remain (it is not known whether others were made): one included in the Pastrana Relation (Guadalajara) and another in that of Consuegra (Toledo). Both of these Spanish towns are located in the centre of the Iberian Peninsula. On the other hand, a very significant number of sketches or maps of the most varied origin remain from America, although it is true that not all the reports were accompanied by maps and that not all of them have been preserved. Some are more intuitive and others more technical; some focus on the space around the locality, roads, mountains, volcanoes, rivers, salt mines, mines, and springs, etc., while others focus on urban aspects, municipalities and larger areas. Additionally, some include texts in local languages and drawings alluding to ancient deities, glyphs among others. Simply put, they are textual and graphic documents that unite both the old and the new. This material captures the changes in the landscape: the union of pre-existing elements and those brought in from outside.

The extremely revealing questions, concerning the settlement and the population and referring to the anthropogenic environment, delve into examining the social structure of a population and the different trades and leave a basic impression of the rural areas. This can be seen, for example, in many of the sketches taken from around the fields, where the newly incorporated livestock and stock species are drawn together with the native trees and vegetation. Some examples of maps or paintings, as they are sometimes called in the sources, give an idea of the types of maps that were made.

Perhaps one of the most interesting maps in terms of landscape and the fusion of cultures is that of Macuilxóchitl and his jurisdiction (Oaxaca) from 1580 (Figure 5), in which elements of Zapotec mythology, Christian elements, annotations in Spanish and Nahuatl, and the lingua franca of Mesoamerica, are mixed. The main glyph, depicting a mountain where three human figures are located: Coqui Pilla (Lord Snake), Ciqui Piziat (Lord Golden Eagle) and Lady Yozi Xonaga Pela Laa (house)-are located, represents the hill Cerro Danush. In addition, the annotations included in the drawing and text of the relation make it possible to identify the elements of the landscape and the legend behind

the different symbols. The roads are identified through the use of symbols of bare feet and horseshoes, the river is identified by the colour blue, the settlements by houses topped by a cross, the livestock are scattered throughout, and the vegetation is depicted as agave and prickly pears in the lowlands, and trees and features in green on the hills.

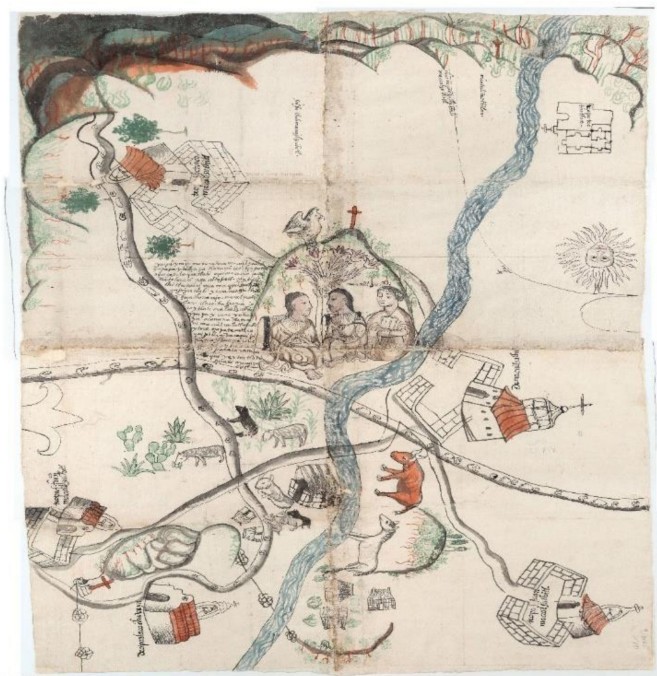

**Figure 5.** Map of San Mateo Macuilxóchitl, provincia de Guaxaca (Oaxaca) Mexico, 1580. Library of Royal Academy of the History of Spain [28]. Reproduced from [or Adapted from] Royal Academy of History Digital Library by the authors of the research (https://bibliotecadigital.rah.es/es/consulta/resultados_ocr.do?id=1595&forma=ficha&tipoResultados=BIB&posicion=1 (accessed on 24 October 2021)). Public Domain Work.

As a technical map, the one of Tlacotalpa (Veracruz), which is practically a nautical chart, is very interesting. It was drawn by the Sevillian sailor Francisco Stroza Gali, a connoisseur of the Mexican coast. According to C. Manso [29] (p. 38), it is likely that Gali was on an official mission when he was entrusted with this work, and that the detailed coastal profiles and the nautical data he provided using the astrolabe were very useful for the Casa de Contratación. In the document, he states that he "has walked and examined all the heights and parts herein contained"; and, at the top centre of the document the following note was included: "Report of the northern latitude in which the land of this description is truly (sic) and faithfully situated". In total, nine localities are mentioned: San Juan de Olua, Punta de Anton Niçardo, Boca de Aluarado, Roca Partida, Tacotalpa, Taliscoya, Tustla, Tlaçinta y Guateupa and Tapacula. The map, dated 1580, also includes a detailed description of the coastline, islands, navigation channels and their depth (bathymetric soundings), rivers, their tributaries and their mouths, lagoons, and human settlements among other features. Additionally, exhibiting a certain degree of technical skill, but of an inland area, the map of Zapotlitán drawn by Juan de Estrada is of particular interest. This chart includes a graphic scale, distances between settlements, and draws and labels the Xaltepec volcano, among other features.

The map of the town of Texupa and its surroundings shows an orthogonal urban layout, formed by 22 square and rectangular blocks, with the square in the centre and the annotation "the town and head of Texupa". Above it is the church, a large stone building with a bell tower, pictured on top. From the square, paths lead off to the east and west, marked using bare footprints. The first path joins up with "the road from Chalco to Tezcuco and from Tezcuco to Chalco" and passes through the farm (estancia) of Santa María. The

second path links up with "the road from Coatlychan to Mexico", with "the road from Tezcuco to Mexico" and with "the road from Mexico to the town of Coatepec", which passes through the farm of Coatongo. Outside the city walls, a toponymic glyph of Texupa is represented: a fountain over which there is a bird (chicuatototl), with the annotation: "the ancient fountain of Texupa from which the town took its name". At the bottom there is a lagoon, with the annotation "the all-blue lagoon", and to the right there are mountain ranges. In the centre of the mountain ranges there is a higher hill, on which a square building has been drawn with the following annotation: "the hill and the house of [...]", "the hill where the Chimalhuacan people worship". The map is dated 1579 (Figure 6).

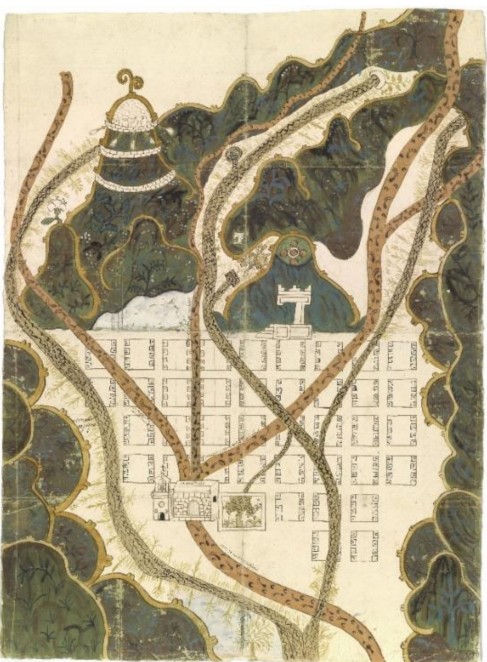

**Figure 6.** Description of the town of Texupa and its jurisdiction, Mexico. 20 October 1579. Library of Royal Academy of the History of Spain [C-028-010] [29]. Reproduced from [or Adapted from] Royal Academy of History Digital Library by the authors of the research (https://bibliotecadigital.rah.es/es/consulta/registro.do?control=RAH20110000400 (accessed on 24 October 2021)). Public Domain Work.

It is important to bear in mind that literal and graphic documentation is complementary when it exists. Additionally, the set of textual information collected in the reports and the annexed graphic information in the maps, paintings or drawings, provides a rather interesting perspective of the people on both sides of the Atlantic. This includes their perception of space, settlement, landscapes, etc., in the second half of the 16th century, the starting point of important changes in the landscape [30] (p. 362).

The documents resulting from the surveys was given two distinct names. With respect to the American set, this was generally referred to as Geographic Relations, while the Spanish documentation was referred to as Topographic Relations. The latter are currently kept at El Escorial library, while the American documents are more dispersed, being housed at the General Archive of the Indies (Seville, Spain), the Royal Academy of History, the University of Glasgow, the Benson Latin American Collection (University of Texas) among other sites. In some cases, the textual and graphic documents are kept in different archives. For example, the Suchitepec Relation (Oaxaca, 1579) is housed at the Royal Academy of History and its map at the General Archive of the Indies. However, in general, when both types of documents exist, they are kept in the same archive [31] (p. 247).

The value of this resource, from a geographical perspective and as an important reference, has already been appreciated in the past, as it was used during the second half of

the 18th century. The Royal Academy of History undertook the creation of the Geographic Dictionary of Spain, entrusting several academics with the task of copying "verbatim" from the Spanish Relations [32] (p. 177). This copy is kept in the archive of the latter and, incidentally, is easier to read than the original source.

## 4. Discussion

In the second half of the 16th century, the traditional approaches for charting territories, such as descriptions and drawings, were considerably enriched through the introduction of the practice and use of questionnaires. This tool allowed the state administration to gain more in-depth knowledge about the territory under its jurisdiction and to use the findings obtained for its own interests and needs. The questionnaires analysed here were designed, executed and used with different visions and objectives, which has an undeniable reflection on the way in which they ask the questions and how they require the answers. In this way, it is necessary to note that the questions of the interrogations pertaining to the topographic relations of Philip II are, for the most part, extraordinarily open in terms of the breadth of data requested and the low precision with which the answers are urged, which undoubtedly promoted very varied information on landscape matters, with few possibilities of generalization and extension for fiscal use. This first major survey exercise served rather as an action aimed at obtaining an instruction on how to proceed in the collection of information about the territory. If the Topographic Relationships, due to their casuistry, remained as a reliable sociological and anthropological analysis of the geographical reality of their time.

The request for, and interpellation of, reports by different levels of government was a constant occurrence in the kingdoms of the Iberian Peninsula during the Modern Age. Specifically, the surveys of Philip II and the method for gathering spatial information designed was not only useful at the time, but also constituted a model and a true reference point for the enlightened of the 18th century. The natural desire for gaining scientific knowledge at a time when research was beginning to form part of the day-to-day organisation of territorial management was instantly united by an economic factor. Since data had been obtained from the various surveys carried out, the States were able to gather information more efficiently about the actual state of the territory and the ways in which the inhabitants were using the resources in the area under survey.

Among the questionnaires designed during this century, following the path laid down two centuries earlier by the relations of Philip II, we find, in Spain, some of the most relevant cadastres-inventories in Europe of the time; that is, the one by Patiño for Catalonia and the Ensenada for the Crown of Castile, as well as that of the geographer Tomas López and the Audience of Extremadura, among others. These questionnaires were designed, executed and used for different outcomes and purposes, which is undeniably reflected in the way the questions and the answers required were formulated. In this way, it is noteworthy that the questions of surveys belonging to the topographical relations of Philip II are, for the most part, extraordinarily open in terms of the breadth of data requested and the fact that the requested answers were not very specific. This undoubtedly encouraged those filling in the questionnaire to provide extensively varied information about the landscape, with little possibility for simplification and or fiscal application. This first major cadastral survey served more as an exercise for obtaining ideas on how to proceed with the collection of information on the territory. If the topographical relations, due to their casuistry, remained a reliable sociological and anthropological analysis of the geographical reality of their time, the other questionnaires dealt with here came to light with a very different intention in mind. The questions contained in the 18th century surveys, which were quite specific in nature and non-anthropological for practical reasons, are indicative of being purely for management purposes, of which the Cadastre of Patiño for the four Catalan provinces is a paradigm.

These types of documentary information sources, of the most varied diversity of periods and contents, have been widely used as a methodological foundation by many

social and natural sciences, although in recent decades the geographical discipline has managed to make optimal use of textual information and cartographic that contain the analysed paracatastral documents, focusing them on some of the most discussed fields of research, such as regional studies of the territory and the analysis of the landscape from its perspective of historical comparison. After reviewing the latest developments in treatment and research with documentary sources, it is clear that geohistorical sources hold a large set of data that, well managed, provide a source of knowledge not only for the structure of the society of the moment in which they were carried out, but also serve as the basis for other types of research. Having analysed the needs of this group of sources, we understand that the most plausible solution is to establish a meeting point for researchers, from which the general public can also benefit. For all this, the development of specific tools was required, the functionalities of which have been on the roadmap for some years: firstly, of the SIGECAH initiative, and currently of the IDE GEOHIS Research Group of the Autonomous University of Madrid.

## 5. Conclusions

The universality and expansion of the paradigms of a knowledge society has supposed, for the vast majority of the sciences, an incentive for adapting their methodologies, procedures, criteria and applications to the demands of the dynamic, changing and competitive world of the 21st century. In the field of Geography, these trends have served as support for the complete and generalised integration of the use of the so-called geotechnologies, which, together with the creation and consumption of large amounts of data with spatial characteristics, has led to a boost in the achievement of technological and research solutions that satisfy the new needs that have arisen. In this context, geohistorical sources are increasingly becoming an inexhaustible source of information and understanding for 21st-century sciences. In the geoinformation society that dominates today's world of culture, science and society, the value of these types of sources is reinforced by the capacity to treasure an incalculable set of heterogeneous data, which can serve as a basis for multiple disciplines, but with a spatial and territorial components as the origin of its contents. Well managed, this volume of data provides, as the medical topographies show, a source of knowledge, not only for the fields of medicine and geography but also about the structure of a society at the time it was created, which constitutes a first-rate collection that can be used as a basis for many different types of research.

After analysing the characteristics and the way of acquiring knowledge of this group of geohistorical sources, the researchers understand that the most notable advantage is the use and application of the cadastral and parachute documentation on Spain in the Enlightenment Century, as a basis for the analysis of the territory. This importance lies in the in-depth study of how and why the analysed documentation was prepared, in order to establish the foundations for a better understanding and use of the information provided to the researcher. In this tidal wave of strong impulses and expansion of the Earth sciences, the European States played a preponderant role, even more so if the enormous incorporation of overseas territories into the maritime empires of the moment is put into perspective.

In the second half of the sixteenth century, the traditional instruments of knowledge of the territory, such as description and representation in drawing, were considerably enriched with the introduction of the practice and use of interrogations, which allowed the State Administration to gain more exhaustive knowledge about the territory under their jurisdiction and the use of the inquiries made in pursuit of their own interests and demands.

**Author Contributions:** Conceptualization: C.C.-B. and A.V.-R.; methodology: C.C.-B., A.V.-R. and M.A.B.-G.; validation: A.V.-R., Á.I.A.-C. and L.G.-J.; formal analysis: C.C.-B., A.V.-R. and Á.I.A.-C.; investigation: C.C.-B. and A.V.-R.; writing—original draft preparation: A.V.-R., M.B.B.-C. and C.C.-B.; writing—review and editing: A.V.-R., M.B.B.-C. and C.C.-B.; project administration, C.C.-B. and A.V.-R. All authors have read and agreed to the published version of the manuscript.

**Funding:** This research was funded by project PID2019-106735 GB-C21, from the call I + D 2019, "Advancing our Understanding of the Land Registry and Other Cadastral Sources: New Perspectives Based on Complementarity, Modelling and Innovation" (Avanzando en el conocimiento del catastro de Ensenada y otras fuentes catastrales: nuevas perspectivas basadas en la complementariedad, la modelización y la innovación), of the Spanish Ministry of Science and Innovation, and the Knowledge Transfer project FUAM-465026 awarded by the Autonomous University of Madrid Foundation (FUAM) and the Spanish General Directorate for Cadastre "New Methods and Approaches for the Transfer in the Social Sciences and Humanities in Cadastral Issues: A Story Worth Telling (Nuevos métodos y enfoques para la transferencia en ciencias sociales y humanidades en materia catastral: una historia que merece ser contada).

**Institutional Review Board Statement:** Not applicable. This study not involving humans or animals.

**Informed Consent Statement:** Not applicable.

**Data Availability Statement:** Not applicable.

**Conflicts of Interest:** The authors declare no conflict of interest. The funders had no role in the design of the study; in the collection, analyses, or interpretation of data; in the writing of the manuscript, or in the decision to publish the results.

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
