# Peer review of "Discovering the Legacy of Hispanic/Spanish and South American Landscapes through Geohistorical Sources: The Geographical and Topographical Relations of Philip II"

_sustainability, doi:10.3390/su14031306_

Round 1

Reviewer 1 Report

The paper deals about interesting topic but must be seriously improved. It brings only the description but not analysis and synthesis.

Introduction: I like the typology of sources and also characteristics of the sources are nice. But I miss mentions how to work with sources by GIS and statistics.

Material and Methods: The general principles how to read sources are clearly described. On the other hand, specific questions focusing on landscape legacy are missing. How authors deal with the sources? Read them? Do GIS analysis? Do reconstruction maps?

Results: All this part descibe the sources in details and should be part of the Materials. Results are missing. How athors discover (reconstruct) the landscape legacy? Present a model area as an example.

Discussion: It mention another sources from different time but also sources from different countries should be mentioned. Are Spanish sources unique in the Europe?

Figures: They are nice but not specifically related to the text.

Title: It is a bit biased to mention the main source title in the title of the paper. Non-Spanish readers are confused.

Author Response

Thank you for your comments and the opportunity to resubmit our manuscript. Please see the attachment.

Reviewer 2 Report

Congratulations to the Authors for taking up an important and interesting topic. The theoretical approach to the issue of geohistorical sources in the reconstruction of landscapes is interesting and inspiring. However, I have some comments on the text.

Technical Notes

1. Working language - caused me a problem. There are many times complex sentences in the text that make it difficult to read and understand. Two examples: (1) "This source has been chosen, due to the quality, quantity, variety and systematicity o the data it provides and because it ended up being the model of how to obtain organized, systematic and homogeneous knowledge of an immense space , without eliminating geographic, anthropological specificities, and historical data of the different territories and because local authorities, both secular and ecclesiastical, are questioned, who are the ones who live, use and, at different levels, govern the territory and its people." (line: 21-26); (2) “Having analyzed the needs of this group of geohistorical sources, the researchers behind this work understand that the most notable advantage, derived from the use and application of the cadastral and paracadastral documentation on Spain in the Age of Enlightenment as a basis for the analysis of the territory, lies in the in-depth study of how and why the documentation was produced in order to establish the fundamentals for the better understanding and use of the information they provide to the researcher. " (line: 421-426).

2. In my opinion, multiple sentences should be avoided in scientific works, if possible. .

3. Record of cited works for correction (line: 34.41; 54; 132; 156; etc.)

4. Map sources cited in the paper should be included in the reference list.

5. No source for “Figure 5. Description of the town of Tlacotlalpa and its jurisdiction, in the bishopric of Tlaxcala, 296 Mexico. February 5, 1580. Stroza Gali, Francisco. Library of Royal Academy of the History of Spain." Line: 296-297. Understands that the "Library of RAHS" information is sufficient for Authors, but need not be for the reader. It is good to include more detailed information. Similarly, Fig. 4. The more so because other figures have signatures (see Fig. 2,3,6).

6. Authors contributions are missing.

7. It seems to me that the entire literature does not comply with the MDPI requirements - https://www.mdpi.com/journal/sustainability/instructions#preparation

8. This article cannot be found [16] “Camarero- Bullón, C. and Aguilar- Cuesta, A.I. (2021). La Cartografía, instrumento para conocer el territorio, planificar y gestionar las reformas en la España del siglo XVIII. Manuscrits. Revista d’Història Moderna, 42.” Has it been released yet?

9. Work [25] does not have a DOI number. https://doi.org/10.1215/00182168-44.3.341; similarly [26] https://doi.org/10.1016/0305-7488(75)90035-3

Remarks to the text

Abstract

1. In my opinion, the abstract does not sufficiently explain the purpose of the research.

2. The abstract also lacks research methods, results and conclusions (parts of the abstract suggested by the MDPI publishing house - see "instructions for authors").

Introduction

1. There is no or too weakly indicated research hypothesis that will be verified in the research results.

2. The aim of the undertaken work is missing or insufficiently indicated. General considerations about the role and importance of geohistorical sources are not enough.

Material and Methods

1. In the "Material and Methods" section, it seems that the research methods used in the work are too weakly emphasized. There is a rationale for the choice of text. A scientific approach is advisable. The role of critical analysis is underlined. And that's probably all - in my opinion, too little for the description of the research method.

2. Does not understand the relationship of the sentence: "Furthermore, this work questions the local authorities, both secular and ecclesiastical, who live in, use and, at different levels, govern the territory and its people." (line: 152-153) with a preceding and following part of the text.

Result

1. The text (line: 160-179) can hardly be considered "Result". This is just a compilation of information that is widely known to historians. So is the text (line: 180-184).

2. I do not see a connection between this text and figure 2. Trinidad is not mentioned in the text. The introduction to "Result" should probably be at the beginning of the text.

3. I do not understand how the placed figure (Figure 3, line: 223) is supposed to be an illustration to the text (line: 215-222). And in the next part of the text, a sudden transition from Venezuela to the border of Castile.

4. Line: 242 "(RAH, Topographical Relations of Philip II)." ?

5. Why is figure 5 (line: 296) quoted in Macuilxóchitl?

6. The text (line: 291-294), if I'm not mistaken, is more about figures no. 6, not no. 5.

Discussion

1. A significant part of the discussion (line: 357-391) has the character of statements resulting from the authors' beliefs (I do not claim to be wrong). However, in the Discussion section, I expect criticism and argumentation, not a statement that this is the case. Additionally, as can be concluded, also from the cited works [Cline, Clinton; 25,26] they are of a secondary nature.

2. They don't bring new knowledge.

3. The discussion of T. Lopez's material does not result from the title or from the earlier parts of the work (line: 392-399).

Conclusions

1. Some of the conclusions do not result from "Result" and "Discussion" (line: 403-419).

2. The last paragraph of the work states what the Authors understand, and not what results from the research.

Research and publication in a journal with a high IF should increase our knowledge. They should have a clearly stated purpose and a verified hypothesis. Unless it's a review article. They should have added value. In this text, the added value, in my understanding, is the collection of general remarks on the importance of geohistorical sources in landscape research on a selected example. Additionally, in the introduction, the author's, in a sense obvious, classification of the sources. The approach to technical issues is also incomprehensible to me - the journal's requirements are clearly stated in the instructions for the Authors. In my opinion, the text still needs a lot of improvement in terms of structure, arguments, discussions and conclusions. Decent technical editing necessary.

Author Response

(The authors gave the same response as above.)

Reviewer 3 Report

The article proposes a geo-historical analysis of some cartographic sources from the age of Philip II. These sources are of great interest for historical cartographic research for aspects only partially highlighted in the text. The authors deal with general and often unsubstantiated questions on the research methodology in historical geography without proposing a methodological path and an analytical framework for sources. Such a generic approach prevents from getting concrete research results. The vacuity of objectives is also evident:
- in the title: I would remove the term 'discovering'; it seems to suggest a descriptive approach rather than an interpretation and analysis of documents;
- in the abstract: it dwells on generic questions that seem to be a premise for the themes addressed in the text that follows; the authors do not indicate a clear and circumstantiate objective of the article, other than that of a generic, undefined geo-historical reconstruction of the territory.
- In the first paragraph (Introduction): the authors dwell on themes that are in themselves quite well known and that can be taken for granted considering the typology of the journal's readers (even if they are not exclusively scholars in historical geography and history of cartography). This aspect is particularly evident in the section on the typology of geohistorical sources. The introduction, on the whole, is improvable because it does not pose research questions and problems that the authors should discuss in the following parts. 
- In section 2 (Materials and Methods): basic methodological questions and introductory issues are again presented.  It would be appropriate to include the description of sources used (specifying which sources, why those, etc.) and the analytical methodology used (what are these sources useful for? Why? Which analytical methodologies authors apply? Why? etc.).
- In paragraph 3 (Results): a series of preliminary questions are proposed again: a historical background which is helpful but not relevant in this section; a description of the sources (to be included in the previous paragraph); a reflection on place names and other aspects of the representation which are too generic. In this section, the reader would expect a precise, detailed analysis of the documents presented in the preceding paragraph and shown in the figures in the light of the methodology used. The authors describe the documents in their fundamental traits, in the elements they highlight, without ever clarifying how they use these aspects to understand and reconstruct "the legacy of landscape". 
The authors allude to the possibility of analyzing the way space is perceived in these documents, but they do not proceed in that sense. Such analysis could represent one of the results of the article. It is emblematic that, in the results paragraph, the authors refer to research to conduct: "This research will briefly refer to toponymy [...]" (line 226). The figures in paragraph 2  in this paragraph should be analyzed, not just described. In this way landscape elements, perceptions of the time, specific interests of their makers, etc. would emerge. It is a pity that such interesting material is only summarily described and not interpreted according to geohistorical methodology. 
- In paragraph 4 (Discussion): the results the authors do not discuss to define and fine-tune the research; they further circumstantiate the context in which develops a kind of state-promoted cartography, without, however, entering into the specific problems encountered, the limits of the results, etc.
- In paragraph 5 (Conclusions): the text suffers from the lack of a specific analysis of the sources; it ends with a sentence stressing the importance of in-depth analysis for 'correct' use of the information contained in sources. That is a relevant consideration, but again, it has the characteristics of a premise rather than a conclusion.  

Salient points with punctual issues:

1.    Lines 30-31: it is not clear what do you mean by terrains
2.    Lines 32: it is not clear what is meant by historical place and society
3.    Line 38: the term paracadastral documents need to be explained, it is not clear enough
4.     Lines 46 and 47: I would reflect on the meaning that the authors attribute to the term landscape, with a reference in the text to the literature examined for this definition. The landscape is considered one of the geographical spaces; the geography of the landscape offers numerous in-depth reflections in this field (see Lucio Gambi, Eugenio Turri, Denis Cosgrove, and more recently Benedetta Castiglioni, among others) that could provide further elements.
5.    Line 62: right sources. What do you mean for right? In what sense? 
6.    Line 68: well managed. See previous
7.    Line 235 more important. See previous
8. 8. Line 249: "popular". Generally, the term popular cartographies (coined by Denis Cosgrove) refers to non-technical cartographies made for a large audience during the 20th century, also for propaganda purposes. Therefore it is worth considering whether it is appropriate to define these cartographies expressing the local perceptions as popular.
9.    Line 298: practically a nautical chart. The nautical chart is a specific genre, born with a purpose, ad usum navigantium. I would soften the expression by speaking of representative features that recall those of nautical charts.

Author Response

(The authors gave the same response as above.)

Round 2

Reviewer 1 Report

The authors added several paragraphs but the main problem of the manuscript remains.

The authors did not specify the research question in the introduction. What is the purpose of the article? The introduction of archival materials is insufficient research aim. Authors should present how they work with these documents and what information about landscape legacy could gain by studying them.

The methods were extenden about paragraph where authors declared that they use hypothetico-deductive method. They wrote that "...the deduced statements are checked or refuted by comparing them with the experience and current state of the territory." But the results do not compare the sources with the current state of the territory but only shortly describe the state in the time when the source was made.

The results also do not support the other statement from methods that "it has become the model to follow for obtaining organised, systematic and homogeneous information about an immense space, without eliminating the geographical, anthropological and historical specificities of the different territories."

The part Results is still the weakest part of the manuscript. It is very descriptive and present selected sources from the big collection of the geographical and topographocal relations of Philip II. The overall characteristic of the geographical and topographocal relations of Philip II is missing. What is the structure of this source? Which territories are covered and which not? How many territories are covered by both map and text report in %?

Only one simple attempt of analysis is made when authors work with toponyms and their origin (case of Segura de la Sierra). Other part of the results is description of selected documents.

There are any references in the discusion!

Authors do not discuss methods how to work with historical sources and their own results.

Use English version of the personal names. So only geographical and topographocal relations of Philip II not Felipe II.

Charles V was emperor in the Holy Roman Empire not in Germany.

I still recomend to modify the title, e.g. Discovering the legacy of Hispanic / Spanish and South American landscapes through geohistorical sources: The geographical and topographical relations of Philip II. The title in the present form is confusing for the non-Spanish reader.

Reviewer 2 Report

I would like to thank the authors for starting work on improving the text and taking into account some of my comments. In my opinion, however, the text still needs to be improved. The comments below correspond to the comments provided by the Authors (blue).

Response to 2 Reviewer Comments

Technical Notes

Point 1: Working language: There are many times complex sentences in the text that make it difficult to read and understand.

Point 2: Multiple sentences should be avoided in scientific works, if possible.

Response 1-2: The language and mode of expression in the two sentences indicated by the reviewer have been modified. Overall, an in-depth review of the text has been done, and it has been re-translated by a professional translation service.

In my opinion, there are still too many multiple sentences. French and Spanish can handle such sentences without a problem. This is not the case in English. If a native speaker translated, I am surprised with the final version.

Point 3: Record of cited works for correction.

Response 3: The works cited in the text have been corrected to adapt all of them to the correct format of the journal.

Not all, see line: 266, still we have “[16-17-18-19-20-21].”

Point 4: Map sources cited in the paper should be included in the reference list.

Response 3: The sources of the figures have been introduced in the bibliography, and the information format has been unified.

Point 5: Record of cited works for correction.

Response 5: All the reviewer's suggestions have been incorporated into the text. The journal's citation regulations have been revised. The sources of the figures have been introduced in the bibliography, and the information format has been unified.

Point 6: Authors contributions are missing.

Response 6: The main contributions of the authors have been introduced, following the criteria of quality and equity of the journal.

It is not done. See Your text, there is not such part like below (from MDPI).

Author Contributions: For research articles with several authors, a short paragraph specifying their individual contributions must be provided. The following statements should be used “Conceptualization, X.X. and Y.Y.; methodology, X.X.; software, X.X.; validation, X.X., Y.Y. and Z.Z.; formal analysis, X.X.; investigation, X.X.; resources, X.X.; data curation, X.X.; writing—original draft preparation, X.X.; writing—review and editing, X.X.; visualization, X.X.; supervision, X.X.; project administration, X.X.; funding acquisition, Y.Y. All authors have read and agreed to the published version of the manuscript.” Please turn to the CRediT taxonomy for the term explanation. Authorship must be limited to those who have contributed substantially to the work reported.

https://www.mdpi.com/journal/sustainability/instructions#preparation

Point 7: It seems to me that the entire literature does not comply with the MDPI requirement.

Response 7: The journal's published criteria regarding bibliographic citations and literature have been reviewed. All references that did not meet the criteria have been adjusted

Changes have been made, sorry to write this, however the literature still does not meet the criteria given by the MDPI. It is enough to check the entry in the revised manuscript and the requirements of the publishing house. I do not understand why to do improvement contrary to the given requirements.

From revised version (e.g. line 728-729):

No. 21 Vallina- Rodríguez, A. and Konyushikhina, N. (2017): Los interrogatorios de los catastros españoles de la Edad Moderna: fuen-728 tes geohistóricas para conocer los paisajes y las sociedades. CT Catastro, 91, pp. 39-63.

Should be according MDPI:

https://www.mdpi.com/journal/sustainability/instructions#preparation

Author 1, A.B.; Author 2, C.D. Title of the article. Abbreviated Journal Name Year, Volume, page range.

Etc., etc. It's a simple notation, I don't understand why it wasn't applied, and I have to write about the technical issues again.

Point 8: This article cannot be found [16] “Camarero- Bullón, C. and Aguilar- Cuesta, A.I. (2021). La Cartografía, instrumento para conocer el territorio, planificar y gestionar las reformas en la España del siglo XVIII. Manuscrits. Revista d’Història Moderna, 42

Response 8: The indicated article is accepted and pending publication by Manuscrits. The editor-in-chief of the journal has been consulted, who has confirmed that in the near future number 42, and the referred article, will already appear published. Possible second rounds of evaluation are expected to be able to assign pagination to the article, or remove it from the bibliography if necessary.

For this moment (10.12.2021) not edited.

Point 9: Works does not have a DOI number.

Response 9: The suggestion of incorporation of these two DOIs is appreciated. They have been added to the mentioned articles.

Abstract

Point 1: The abstract does not sufficiently explain the purpose of the research.

Response 1: Based on the reviewer's comments, the abstract has been reformulated to better explain the starting hypothesis, the research method with geohistorical sources and the results.

In my opinion, the abstract still does not sufficiently explain the purpose of the research undertaken. I kindly ask you to format the abstract according to the MDPI formula.

Abstract: A single paragraph of about 200 words maximum. For research articles, abstracts should give a pertinent overview of the work. We strongly encourage authors to use the following style of structured abstracts, but without headings: (1) Background: Place the question addressed in a broad context and highlight the purpose of the study; (2) Methods: briefly describe the main methods or treatments applied; (3) Results: summarize the article's main findings; (4) Conclusions: indicate the main conclusions or interpretations. The abstract should be an objective representation of the article and it must not contain results that are not presented and substantiated in the main text and should not exaggerate the main conclusions.

Point 2: The abstract also lacks research methods, results and conclusions.

Response 2: Based on the reviewer's comments, the abstract has been reformulated to better explain the starting hypothesis, the research method with geohistorical sources and the results.

In the Authors' response, there is still no reference to the request for the “Introduction section”. So I repeat: “Introduction. There is no or too weakly indicated research hypothesis that will be verified in the research results. The aim of the undertaken work is missing or insufficiently indicated. General considerations about the role and importance of geohistorical sources are not enough.”

Why the given hypothesis (line: 211-212) is in the part “Material and Methods”? Cit. "fundamental hypothesis that the landscape and the territory have a clear reference of knowledge in geohistorical sources" is in a sense a tautology in relation to maps and censuses, even in this historical period. The study proved that the analyzed material, text and maps, which were prepared to describe the existing state of affairs, describe (more or less precisely) this landscape. And we already know this since the creation of "Topographic Relationships of Felipe II", which is also referenced in many works by earlier authors. What I wrote about in the previous review.

Material and Methods

Point 1: In the "Material and Methods" section, it seems that the research methods used in the work are too weakly emphasized.

Response 1: Additional information on the methodology used and the direction of research that has been followed has been included in the manuscript.

In the part “Material and Methods” we have new part (line 212-218).

However, this part "From this, the phenomena included in the geographical descriptions are explained, the most elementary consequences or implications of the hypothesis itself are deduced and, finally, the deduced statements are checked or refuted by comparing them with the experience and current state of the territory" (line 215-218) appears to be wrong. The correctness of the description / presentation of the landscape from sources from the 16th century cannot be stated (verified or refuted) compared to the present state in the 21st century. For 500 years various anthropogenic and natural processes have taken place, the landscape has changed, if only because of the territorial development of cities and the enlargement of their administrative borders.

Point 2: Does not understand the relationship of the sentence: "Furthermore, this work questions the local authorities, both secular and ecclesiastical, who live in, use and, at different levels, govern the territory and its people." (line: 152-153) with a preceding and following part of the text.

Response 2: The materials and methods section has been reconfigured, and the phrase mentioned by the reviewer has been removed from the context.

Result

Point 1-3: Do not see a connection between this text and figures.

Response 1: The reviewer's constructive comment is appreciated. The necessary changes are made to provide greater coherence to the text-figures relationship.

Point 4: Line: 242 "(RAH, Topographical Relations of Philip II)." ?

Response 4: Removed this phrase.

Point 5-6: Do not see a connection between this text and figures

Response 5-6: The reviewer's constructive comment is appreciated. The necessary changes are made to provide greater coherence to the text-figures relationship.

Discussion

Point 1-3: A significant part of the discussion has the character of statements resulting from the authors' beliefs

Response 1-3: The reviewer's comments have been helpful in improving this discussion section. The section has been reformulated almost completely, to give it a greater argumentation based on research experience.

In the discussion, the "Result" received should be compared with other research results. Confirm / deny, show new directions. There is no such statement here. The authors write about their opinions and expectations. It is not, in my opinion, not a discussion, but a presentation of the authors' opinions. Not a single work devoted to the subject of research was cited in the Discussion.

Conclusions

Point 1-3: The last paragraph of the work states what the Authors understand, and not what results from the research.

Response 1-3 : Despite not agreeing with some of the content stated by the reviewer, we proceed to a remodeling of the conclusions section.

In my opinion, the introduced changes are still insufficient. The text Conclusion does not fully reflect the results of the research undertaken. The following items: (line: 633-649) not tested, but in conclusion; (line 656-657) not tested but in conclusion; (line: 661-664) not tested but in conclusion.

Reviewer 3 Report

It should be noted that the commentary on point 1, as well as others, was drafted with the aim of highlighting certain aspects of the article which, in the opinion of the reviewer, are open to improvement. It is for this reason that I have pointed out that the fruitful research activity of the group has not been well exploited in the previous version of the text. In spite of some of my notes that were judged irrelevant and useless (which I regret very much), it seems clear to me that the review has contributed to the process of improving the paper. 

Author Response

Thank you very much for your review. 

Round 3

Reviewer 1 Report

Any other comments.

Author Response

As already stated in the two previous review rounds, the authors consider that the author of this review did not understand, from the beginning, the meaning and approach of this manuscript. The effort made to improve the text in the revision rounds is really appreciated.

Reviewer 2 Report

I appreciate the efforts and work of the Authors on the text that has been partially corrected. Below are some comments on the Authors' answers.

Response to 2 Reviewer Comments

Technical Notes

Point 1: In my opinion, there are still too many multiple sentences. French and Spanish can handle such sentences without a problem. This is not the case in English. If a native speaker translated, I am surprised with the final version.

Response 1: We especially appreciate the thoroughness of the reviewer, and his effort to improve the quality of the text. Regarding questions about the translation, we have again consulted the official language service that translated the manuscript. They have indicated that they consider the translation adequate in formal terms and that they consider the work submitted as good.

It is true that we observe the reviser's statements as adequate and correct, but we are not technically or financially capable of dealing with another official translation.

I understand the meaning of the problem, some native speaker of mine could help in this regard.

Point 2: Record of cited works for correction. Not all, see line: 266, still we have “[16-17-18- 19-20-21].”

Response 2: The necessary information has been added to the registry of works cited. Although reading the publication rules for authors does not make it clear whether the rules for multiple citation should follow the same format as single citations, the change has been made as the reviewer warned.

Thanks for the information and explanation.

Point 3: Authors contributions are missing.

Response 3: Information about the authorship and contribution of each author entered. Sorry for the mistake.

Just for the record, I would like to ask what the validation was about in this text. It is possible that I understand this concept differently.

Point 4: It seems to me that the entire literature does not comply with the MDPI requirement.

Response 4: The bibliography has been revised. Thank you for including the link to the guidelines.

I do not understand why after References we have a job as below. If it was a template for writing literature, it should be deleted.

(see line: 501-501) Bowman, C.M.; Landee, F.A.; Reslock, M.A. Chemically Oriented Storage and Retrieval System. 1. Storage and 501 Verification of Structural Information. J. Chem. Doc. 1967, 7, 43-47; DOI:10.1021/c160024a013.

I am a bit tired of the revision of the way literature was written for this work. It is not difficult. Please visit the Sustainability page. Find any published article. Check how the authors wrote the references. And do it correctly. I sent the link previously. There is also information on how to do. It's simple, just stick to the given formula. I understand that these are just technical issues. However, I believe that since the requirements (MDPI) are given, they should be met. Even if it's just technical stuff. Only in items no. 3-5 of the "revised" literature I found 10 errors (full dot, commas, missing information required) in the notation.

I also do not understand why the previous versions have the headline "Sustainability" and the last version is from "Diversity".

Point 5: This article cannot be found [16] “Camarero- Bullón, C. and Aguilar- Cuesta, A.I. (2021). La Cartografía, instrumento para conocer el territorio, planificar y gestionar las reformas en la España del siglo XVIII. Manuscrits. Revista d’Història Moderna, 42

Response 5: You choose to withdraw the indicated article, waiting for it to be included in the publication process

So far (11 Jan. 2022) not released.

 Abstract

Point 1: The abstract does not sufficiently explain the purpose of the research.

Response 1: The abstract has already been reformulated and follows the MDPI parameters.

In my opinion, only changes were made to the way the text was written.

Point 2: There is no or too weakly indicated research hypothesis that will be verified in the research results. The aim of the undertaken work is missing or insufficiently indicated. General considerations about the role and importance of geohistorical sources are not enough

Response 2: Based on the reviewer's comments, the abstract has been reformulated to better explain the starting hypothesis, the research method with geohistorical sources and the results.

Thank You for changes.

Material and Methods

Point 1: The correctness of the description / presentation of the landscape from sources from the 16th century cannot be stated (verified or refuted) compared to the present state in the 21st century. For 500 years various anthropogenic and natural processes have taken place, the landscape has changed, if only because of the territorial development of cities and the enlargement of their administrative borders.

Response 1: Additional information on the methodology used and the direction of research that has been followed has been included in the manuscript.

I appreciate the extension of the text. But I still do not know what hypothesis the Authors are referring to in the given phrase. The title of the text lacks the formulation of the research hypothesis.

(see line: 155-157) "From this, the phenomena included in the geographical descriptions are explained, the 155 most elementary consequences or implications of the hypothesis itself are deduced and, finally, the deduced statements are checked or refuted by comparing them with the experience and current state of the territory. "

Discussion

Point 1: In the discussion, the "Result" received should be compared with other research results. Confirm / deny, show new directions. There is no such statement here. The authors write about their opinions and expectations. It is not, in my opinion, not a discussion, but a presentation of the authors' opinions. Not a single work devoted to the subject of research was cited in the Discussion.

Response 1: The reviewer's comments have been helpful in improving this discussion section. The section has been reformulated almost completely, to give it a greater argumentation based on research experience.

Thank you for expanding the text. For effort and work. As he understands, it is impossible to conduct a scientific discussion of the results achieved with the results of other studies. They could confirm or contradict the results achieved by the authors. The current - corrected discussion only enlarges the description. Unfortunately, it still does not meet the features of a scientific discussion of the obtained research results.

Conclusions

Point 1: In my opinion, the introduced changes are still insufficient. The text Conclusion does not fully reflect the results of the research undertaken. The following items: (line: 633-649) not tested, but in conclusion; (line 656-657) not tested but in conclusion; (line: 661-664) not tested but in conclusion.

Response 1: Despite not agreeing with some of the content stated by the reviewer, we proceed to a remodeling of the conclusions section.

Despite the expansion of the text - appreciating the efforts of the Authors, I will stick to my previous position.

Extra note. Please check it, I cannot find the reference to figure 5 in the text (last version signed as “Diversity”).

Author Response

Thank you very much for the enormous effort made with this text. It has improved in a very clear way. Please, review the response document that we sent attached
